# Transcriptome-wide identification of novel circular RNAs in soybean in response to low-phosphorus stress

**Lingling Lv[1], Kaiye Yu[1], Haiyan Lü[1], Xiangqian Zhang[1], Xiaoqian Liu[1], Chongyuan Sun[1], Huanqing Xu[1], Jinyu Zhang[2], Xiaohui He[3]\*, Dan Zhang** **[1]\***

**1** Collaborative Innovation Center of Henan Grain Crops, College of Agronomy, Henan Agricultural University, Zhengzhou, China, **2** Collaborative Innovation Center of Modern Biological Breeding, Henan Institute of Science and Technology, Xinxiang, China, **3** Smart City Institute, Zhengzhou University, Zhengzhou, China

\* zhangd@henau.edu.cn (DZ); hexh@zzu.edu.cn (XH)

## Abstract

Low-phosphorus (LP) stress is a major factor limiting the growth and yield of soybean. Circular RNAs (circRNAs) are novel noncoding RNAs that play a crucial role in plant responses to abiotic stress. However, how LP stress mediates the biogenesis of circRNAs in soybean remains unclear. Here, to explore the response mechanisms of circRNAs to LP stress, the roots of two representative soybean genotypes with different P-use efficiency, Bogao (a LP-sensitive genotype) and Nannong 94156 (a LP-tolerant genotype), were used for the construction of RNA sequencing (RNA-seq) libraries and circRNA identification. In total, 371 novel circRNA candidates, including 120 significantly differentially expressed (DE) circRNAs, were identified across different P levels and genotypes. More DE circRNAs were significantly regulated by LP stress in Bogao than in NN94156, suggesting that the tolerant genotype was less affected by LP stress than the sensitive genotype was; in other words, NN94156 may have a better ability to maintain P homeostasis under LP stress. Moreover, a positive correlation was observed between the expression patterns of P stress-induced circRNAs and their circRNA-host genes. Gene Ontology (GO) enrichment analysis of these circRNA-host genes and microRNA (miRNA)-targeted genes indicated that these DE circRNAs were involved mainly in defense responses, ADP binding, nucleoside binding, organic substance catabolic processes, oxidoreductase activity, and signal transduction. Together, our results revealed that LP stress can significantly alter the genome-wide profiles of circRNAs and indicated that the regulation of circRNAs was both genotype and environment specific in response to LP stress. LP-induced circRNAs might provide a rich resource for LP-responsive circRNA candidates for future studies.

**Data Availability Statement:** All relevant data are available in the paper, its Supporting Information files, and from the NCBI BioProject, SRA accession number SRP233239.

**Funding:** This work was funded by the key scientific and technological project of Henan Province (192102110023 and 30601916) to DZ, the Key Scientific Research Projects of Higher Education Institutions in Henan Province (20A210017) to DZ, the Henan agricultural university science and technology innovation fund (KJCX2019C02) to DZ, the Innovative Research Group Project of the National Natural Science Foundation of China (CN) (31701309) to HL, and the Ministry of Science and Technology of China (2016YFD0100500). The funders had no role in study design, data collection and analysis, decision to publish, or preparation of the manuscript.

**Competing interests:** The authors have declared that no competing interests exist.

## Introduction

Phosphorus (P), a macronutrient present in all living cells, is essential for plant growth and development because of its functions in energy transfer, in DNA in terms of inheritance, in membrane components, in carbon metabolism and in regulation and signaling [1]. The low availability of soil P and the acquisition of P exclusively through plant roots result in P being a major factor limiting plant growth and development. The sustainability of the use of P fertilizers to optimize crop yields is in jeopardy due to the crisis of the global reserve of rock phosphate and P pollution. Compared with other crop species, soybean needs more P for growth and development; thus, low-P (LP) stress has become a major factor limiting soybean production [2]. Therefore, elucidating the response and adaptation mechanism of soybean to LP stress is of great urgency and importance.

Plant growth and crop productivity are greatly affected by various environmental factors. As such, plants have evolved sophisticated adaptation mechanisms for sensing and responding to suboptimal environmental stresses. Recent studies have revealed that a vast amount of noncoding RNAs (ncRNAs), for example, small interfering RNA (siRNA), microRNA (miRNA), and long noncoding RNA (lncRNA), are induced during abiotic stress, including LP stress [3], suggesting their indispensable role in regulating plant defense responses. In *Arabidopsis*, miRNA can regulate phosphate homeostasis [4]. Changes in miRNA in leaves and roots constitute an important mechanism in maize to adapt to low-phosphate environments [5]. Phosphate starvation also induces miRNA responses in *Populus tomentosa* [6]. Moreover, as a long-distance signal, miR399 can regulate phosphate homeostasis in plants [7] and can serve as a potential integrator of photoresponses and phosphate homeostasis [8, 9]. In *Arabidopsis*, a new class of small RNAs has been identified in phosphate-starved roots [10], and these small RNA mediate responses to P deficiency and regulate P homeostasis [11]. In yeast, lncRNA can dynamically regulate *pho1* expression by recruiting RNAi and the exosome on the basis of phosphate levels [12]. Moreover, lncRNA not only governs the expression of the phosphate transporter gene *Pho84* but also exerts linkage effects on prt lncRNA and *pho1* genes in flanking regions [13]. Recent studies have shown that lncRNAs exert a non-cell-autonomous response to P deficiency and may act as systemic signaling agents at the whole-plant level to coordinate early P deficiency signals [14]. LncRNAs also play key roles in regulating mRNA levels of a large number of genes associated with P starvation responses [15]. Moreover, in the legume model plant *Medicago truncatula*, novel P deficiency-induced lncRNAs can regulate P transport and P-deficiency signaling, suggesting that lncRNAs play an important role in regulating plant responses to LP stress [16].

Recently, a newly characterized type of endogenous ncRNA, circular RNA (circRNA), has been intensively studied in a variety of species [17]. Unlike traditional linear RNAs, circRNAs are generated from head-to-tail back-splicing; thus, they do not have the terminal 5′ cap and the 3′ polyadenylated tail structures found in linear genes [18]. Increasing amounts of evidence have indicated that circRNAs are commonly produced by thousands of genes and have substantial functions in regulating gene expression at multiple levels in eukaryotic cells [19]. For example, circRNAs can function as transcriptional and posttranscriptional regulators, miRNA sponges, small nucleolar RNAs (snoRNAs), RNA processing intermediates, and transcriptional cis-regulators [19–21]. Moreover, increasing amounts of evidence have indicated that circRNAs play a key role in plant responses to abiotic stress [20, 22–26]. For example, circRNAs are involved in plant dehydration and immune responses [26], chilling stress [27], heat stress [20], dehydration [28], infection by pathogenic bacteria [19, 21], and LP stress [29]. These findings indicate that circRNAs are abundant in plants and have important functions in

response to abiotic stresses. However, circRNAs have rarely been reported in soybean [30]; furthermore, no studies have been reported on the response of soybean circRNAs to LP stress.

In this study, to quantify circRNAs in soybean and explore their potential functions in regulating the response to LP stress, we first treated two representative soybean genotypes that present contrasting P tolerances, Bogao (B) and NN94156 (sensitive and tolerant genotypes to LP stress, respectively), with two different supplies of P: LP (-P, 5 μM) and high P (HP) (+P, control, 500 μM). Using genome-wide high-throughput RNA sequencing (RNA-seq) technology, we then identified all circRNAs in the roots of soybean seedlings and validated them by real-time PCR. Our study specifically (1) identified and characterized soybean root circRNAs that are responsive to LP stress; (2) identified and characterized the possible roles of the circRNA-host genes of differentially expressed (DE) circRNAs in regulating soybean tolerance to LP stress via Gene Ontology (GO) enrichment analysis; and (3) predicted and discussed the sponge action of DE circRNAs and miRNA-targeted genes.

## Materials and methods

### Plant materials and hydroponic experiments

The LP-tolerant genotype Nannong 94156 (NN94156) and LP-sensitive genotype B were grown hydroponically as described previously [2]. Briefly, the seeds were germinated and grown in an artificial intelligence climate chamber under 10 h light/14 h dark and 28/20˚C. When the two cotyledons had fully expanded, the soybean seedlings were transplanted into modified half-strength Hoagland's nutrient solution (pH = 5.8, 500 μM Pi, $KH_2PO_4$, sufficient P, HP). After three days, 1/2 of the seedlings were transferred to Hoagland's nutrient solution that contained a low supply of P (5 μM P, P deficiency, LP), with the other 1/2 half the seeds remaining in the P-deficient conditions as controls. The solution was replenished every 3 d, and the soybean plants were grown in a completely randomized block design. Root tissues were collected from the treatment and control plants at 10 d after the plants were transferred to the P-deficient conditions, after which the tissues were stored at −80˚C. Three independent biological replicates per genotype, each from which 12 samples were collected (HP-NR-1, 2, and 3; LP-NR-1, 2, and 3; HP-BR-1, 2, and 3; and LP-BR-1, 2, and 3), were used.

### Library construction and Illumina sequencing

Total RNA from the 12 samples was extracted by Trizol (Life Technologies Inc., USA), and the quality of the RNA was verified via 1% RNase-free agarose gel electrophoresis. A Nanodrop 2000 device (Thermo-Fisher Scientific, USA) was used to quantify the RNA purity, and the RNA integrity was determined by a 2100 Bioanalyzer (Agilent Technologies, USA). First, to obtain rRNA-free residue, an Epicentre Ribo-zero rRNA Kit (Epicentre, USA, cat: MRZSR1 16) was used to remove the rRNA. Second, according to the instruction manual provided by Gene Denovo Biotechnology Co. (Guangzhou, China), RNA libraries that were highly strand specific were generated by a NEBNext Ultra Directional RNA Library Prep Kit for Illumina (NEB, USA) in conjunction with 1 μg of rRNA-depleted RNA. In brief, divalent cations were used to fragment the rRNA-depleted RNA under elevated temperature in NEBNext Reaction Buffer. First-strand cDNA was then synthesized by reverse transcriptase and random hexamer primers. Second-strand cDNA was subsequently synthesized by DNA polymerase I, second-strand synthesis reaction buffer, dUTP, and RNase H. The double-stranded cDNA was then purified and collected via AMPure XP beads for the subsequent steps. Afterward, to prepare for hybridization, the 3′ ends of the DNA fragments were repaired, and the hairpin loop structures were ligated to the fragments. To select cDNA fragments that were 200 to 500 bp in length, the fragments in each of the libraries were purified by an AMPure XP bead system. The

second-strand cDNA, which contained uracil (dUTP), was digested using 3 μL of NEBNext USER Enzyme at 37°C for 15 min followed by 30 s at 98°C before PCR. Quantitative polymerase chain reaction (qPCR) was subsequently conducted in conjunction with universal PCR primers, NEBNext High-Fidelity PCR Master Mix, and Index (X) primer. To assess the library quality, an Agilent Bioanalyzer 2100 system was used to purify the PCR products (AMPure XP system). The qualified libraries were then constructed and sequenced on an Illumina HiSeq 4000 platform, and long paired-end reads (150 bp) were generated.

### RNA-seq and identification of circRNAs

In this study, the FastQC program was used to check the quality of the raw sequencing reads [31], and quality control was performed using the Trimmomatic v0.30 program [32]. The reads were filtered according to three criteria: (1) reads containing adapters were removed; (2) reads containing more than 10% unknown nucleotides (nt) were removed; and (3) reads with more than 50% low-quality (Q-value ≤ 20) bases were removed. These filtered reads were subsequently aligned to the soybean reference genome, Williams 82 *Wm82.a2.v1*, by TopHat2 [33]. The unmapped reads were then collected and analyzed with find_circ [17] to identify circRNAs, which were annotated by searching against the circBase according to the E value via BLAST [34]. With respect to database comparisons, an E value < e-10 was defined for the existing circRNAs and, conversely, for the novel predicted circRNAs; as a result, DE circRNAs were identified between two comparison groups when the fold change (FC) was ≥ 2 ($P < 0.05$). Enrichment analysis of GO functions of the parent genes of DE circRNAs was performed via the GO term enrichment online tool at SoyBase (https://soybase.org/). A heatmap of the results was created using the heatmap.2 function in the gplots package [35]. When the genomic DNA sequence of annotated genes was uniquely aligned with the circRNAs containing head-to-tail splicing sites, the circRNAs were determined to originate from the annotated genes, which we refer to as the circRNA-host genes of the circRNAs. In addition, if circRNAs originated from a fragment between two genes, we considered that the circRNAs had no circRNA-host genes and are defined as "NA".

### Quantitative real-time PCR (qRT-PCR)

In this study, qRT-PCR was used to further validate the circRNA expression levels identified by RNA-seq. Thirteen randomly DE circRNAs, including nine circRNAs predicted to act as miRNA sponges, were selected for experimental validation, and the sequences of the primers used are listed in S2 Table. To maintain the reliability of the experiment, we used the same RNA samples for the transcriptome sequencing analysis to verify the RNA-seq results. Briefly, each PCR mixture consisted of real-time PCR SYBR MIX (10 μL, Toyobo, USA), first-strand cDNA (50 ng), and gene-specific primers (0.5 μL of 10 μmol L$^{-1}$). The PCR amplification procedure used in this study was as follows: 95°C for 5 min, followed by 40 cycles of 95°C for 15 s and then 60°C for 60 s. The soybean internal control gene *TUBULIN* (GenBank accession: AY907703) was used as a control, and the cDNA template was replaced by ddH$_2$O as a negative control. The relative gene expression data were analyzed using the $2^{-\triangle\triangle CT}$ method [36], and three biological and technical replicates were included.

### Analysis of miRNA sponges

For novel circRNAs, the software Patmatch (v1.2) (https://academic.oup.com/nar/article/33/suppl_ 2/W262/2505454) was used to predict target miRNAs within plant samples and to predict mRNAs targeted by miRNA sponges and the interaction between the mRNAs and miRNA.

## Results and discussion

### Identification and characteristics of circRNAs responsive to different P levels

Recently, although the involvement of circRNAs in plant tolerance to abiotic stress has been reported in several species [19–21, 26], their role has rarely been reported in soybean [30], especially under LP stress. In the present study, to identify and explore how circRNAs respond to LP stress in soybean, root samples of two representative soybean genotypes, NN94156 and Bogao, which present significant differences in their traits related to P-use efficiency, were used to construct RNA-seq libraries under LP (5 μM, -P) and HP (500 μM, +P, control) conditions. In total, 12 root samples, including four samples (the roots of NN94156 (NR) and the roots of Bogao (BR) plants under two P treatments, HP and LP) with three biological replicates, were collected and sequenced. Transcriptome sequencing generated approximately 1087 million raw paired-end reads from the two soybean genotypes and two different P levels, ranging from 75.5 to 100.7 million reads per sample. After quality control was performed, more than 90% of reads with high-quality scores (Q ≥ 20) were retained and uniquely mapped to the soybean reference genome sequence (Wm82.a2.v1). The unmapped reads were then collected and processed via find_circ [17] to identify circRNAs. After this two-step analysis, a total of 371 candidate circRNAs were found, which we named novel_circ_000001 to novel_-circ_000371, in these 12 samples; these candidate circRNAs were selected on the basis of at least two unique back-spliced reads and at least two biological replicate samples (S1 Table). Among these circRNAs, 300, 324, 308, and 262 were identified for HP-NR, LP-NR, HP-BR and LP-BR, respectively, across both soybean genotypes under the two different P-supply levels (Fig 1A).

To analyze the types of the 371 circRNAs, we compared the circRNA sequence with the genome origination. The results revealed that six types could be classified: annot_exons, antisense, exon_intron, intergenic, intronic, and one_exon, which contained 47, 144, 112, 23, 5 and 40 circRNAs, respectively (Fig 1B, S1 Table). The 144 antisense circRNAs accounted for 38.8% of the total, and 199 (53.6%) circRNAs contained an exon. These results are consistent with the findings in other species, such as 50.5% of circRNAs containing exon sequences in *Arabidopsis* and 85.7% in rice [29], indicating that the exon_intron type is the most common in all the types and that back-spliced exons are important for the generation of circRNAs. In addition, 204 (55%) circRNAs, including four types, intronic, annot_exons, exon_intron, and one_exon, originated from annotated protein-coding genes (Fig 1B, S1 Table). Moreover, we found that 147 of the 225 circRNA-host genes produced two or more circRNAs (S1 Table), suggesting that the same gene can produce multiple circRNAs by alternative splicing. The data of previous studies, as well as those presented here, demonstrate that a close relationship exists between the origination mechanism of circRNAs and the splicing mechanism of precursor mRNAs [37, 38].

Previous studies have shown that circRNA-host genes are unevenly distributed across different chromosomes [19]. In the present study, we also found that the circRNA-host genes were unevenly distributed across chromosomes (Fig 1C); for example, chromosome 9 contained the most (58) circRNA-host genes, followed by chromosomes 13 and 16, which contained 41 and 30 circRNA-host genes, respectively (Fig 1C). The lengths of the circRNAs ranged from approximately 90 nt to approximately 100 thousand nt, and their host gene distribution across chromosomes was also uneven (S1 Table). The mean length of the one_exon, intronic, annot_exons, intergenic, exon_intron, and antisense circRNAs was 361, 418, 574, 1409, 24607 and 7939 nt, respectively (S1 Fig., S1 Table). With respect to the length distribution of circRNAs, we found that all intronic and one_exon, most annot_exons and some

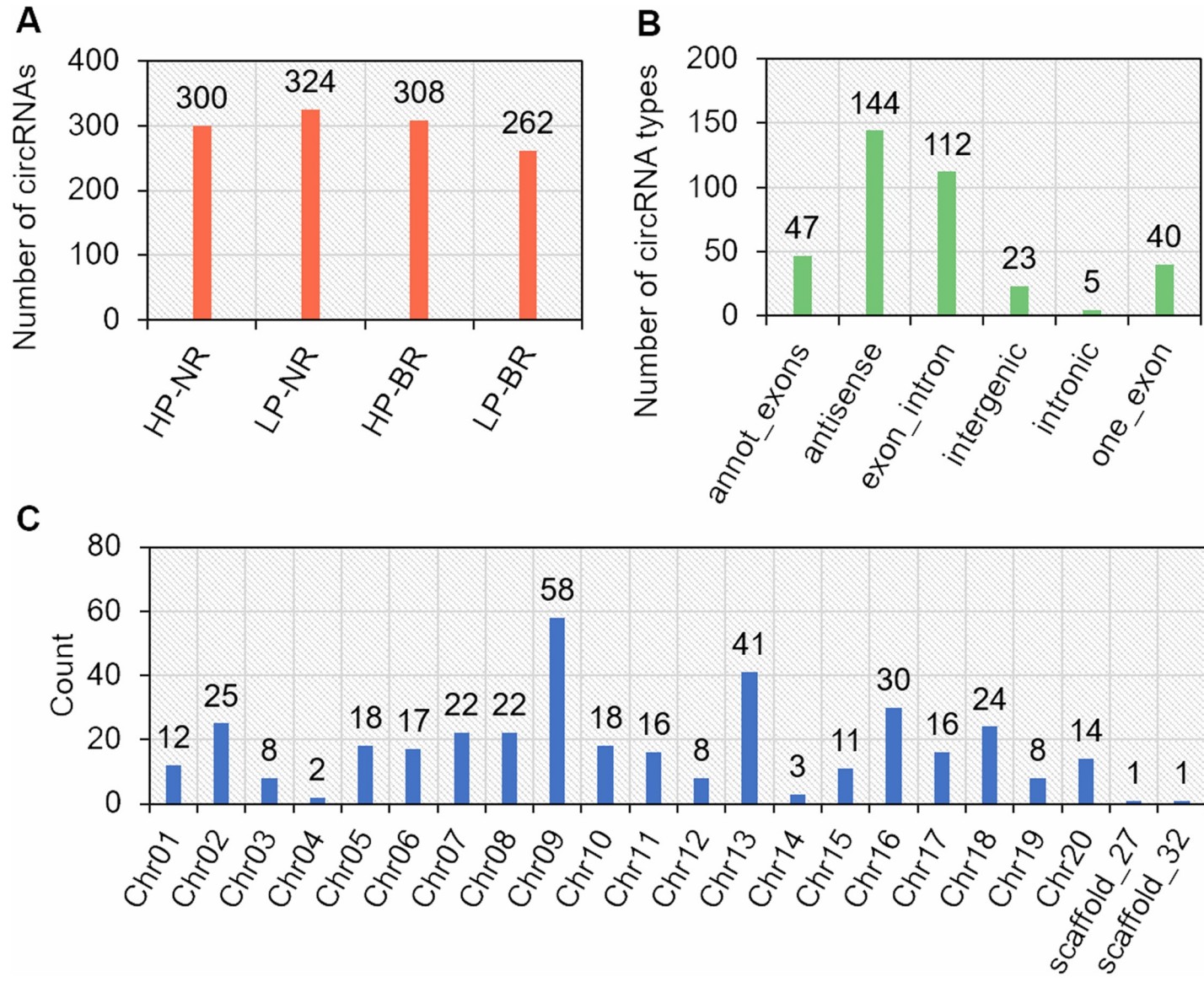

**Fig 1. Identification and characterization of circRNAs in soybean roots.** (A) Histogram of circRNAs identified in each sample under different P levels. (B) Numbers of circRNAs in each type. (C) Numbers of circRNAs from different chromosomes.

antisense circRNAs were shorter than 2000 nt, whereas the majority of exon_intron and a small number of antisense and intergenic circRNAs were longer than 2000 nt (S1 Fig., S1 Table).

The number of circRNAs (371) identified in soybean roots was much lower in our study than in previous studies of model plant species and in studies that analyzed publicly available RNA-seq data; for example, several thousand circRNAs were reported to be present in *Arabidopsis*, rice, cucumber and soybean [29, 30, 39]. The main reason for these differences may be attributed to developmental-, tissue-, or LP stress-specific expression patterns of circRNAs [40]. For example, researchers have recently identified 776, 3171 and 2165 circRNAs in soybean leaves, roots and stems, respectively [30]. Similarly, Zhu et al. [39] identified 2787 circRNAs in cucumber, including 827 in the leaves and 2420 in the roots in response to salt

stress. Thus, some of the circRNAs identified in our study might be root- or P supply-specific, although additional studies are needed. Different sequencing platforms and bioinformatics approaches might also affect the number of circRNAs detected. In cotton, 280 DE circRNAs were isolated on the basis of RNA-seq under conditions of Verticillium wilt stress [21]. Eighty-eight circRNA candidates were identified in wheat seedling leaves, and only 62 were DE under dehydration stress conditions compared with well-watered control conditions [28]. In this study, we used the RNA Ribo-minus RNA-seq method. Our results included mRNA, lncRNA, and circRNA data; however, the expression of circRNA in the examined tissues was extremely low, which may also have caused the small number of identified circRNAs.

## circRNAs are genotype specific and LP stress specific in soybean roots

circRNAs whose expression is genotype specific and environment specific have been detected in plants in response to different abiotic stresses [20, 21]. We performed a Venn diagram analysis to elucidate the relationships among the circRNAs identified. Despite variation in the number of circRNAs across the four samples, a high percentage of common circRNAs (190) was shared among the four conditions, e.g., 63.3% in HP-NR, 58.6% in LP-NR, 61.7% in HP-BR, and 52.5% in LP-BR (Fig 2A, S1 Table). These circRNAs were expressed regardless of the genotype and P-supply conditions, indicating that these common responsive circRNAs are universally present in soybean roots and might be essential to root development. Notably, more circRNAs were induced in NN94156 under LP stress conditions than under the control conditions (HP), the majority of whose expression decreased in the sensitive genotype (Bogao) (Fig 1B, S1 Table). Moreover, the remaining circRNAs were determined to be genotype specific or LP-responsive specific. Briefly, 16 and 40 circRNAs were expressed specifically in NN94156 under HP conditions, and 67 and 21 were expressed specifically in Bogao under LP conditions, suggesting that, in Bogao, more circRNAs were expressed specifically under HP conditions, while in NN94156, more circRNAs were expressed specifically under LP

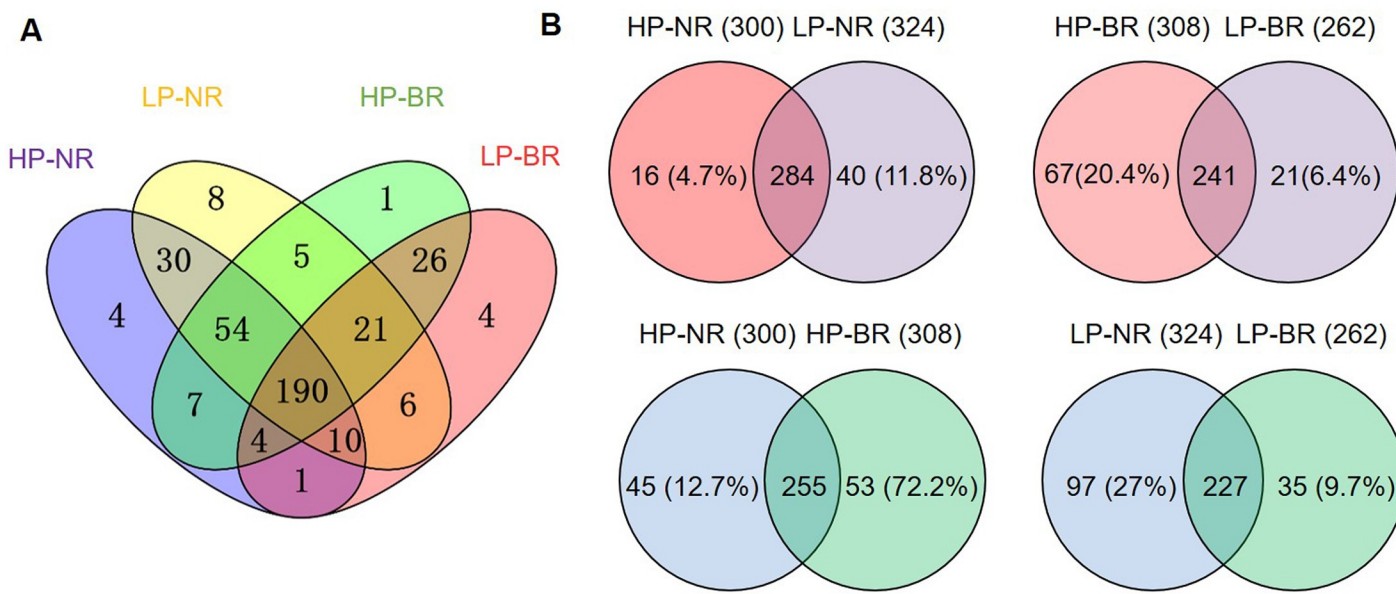

**Fig 2. Comparison of soybean circRNAs in each sample under different P levels.** (A) Venn diagrams of circRNAs identified in the four samples, including the roots of two representative soybean genotypes under different P levels. (B) Venn diagrams comparing expressed circRNAs in each root sample under different P levels.

conditions (Fig 2B). These results indicate that circRNAs may play a crucial role in soybean growth and development. In our study, the common responsive circRNAs among the four conditions might be important to ensure normal root growth, and circRNAs whose expression can be induced in both genotypes under P stress are P stress specific and could be critical for basal defense against environmental stress. Genotype-specific circRNAs have been identified in other species in response to Verticillium wilt stress [21]; thus, the roles of the genotype-specific circRNAs identified in our study merit further exploration and experimental verification.

## LP stress enhances the accumulation of DE circRNAs

Soybean is an important oil-bearing legume with high nutritional value [41]. Compared with the other nonleguminous species, soybean needs greater amounts of P for its growth and development, which is reflected by the greater P content in soybean seeds compared with rice, maize and wheat seeds [2]. Therefore, P deficiency is a more serious problem than other nutrient deficiencies in soybean [42]. In this study, to understand thoroughly how these circRNAs respond to LP stress, we investigated their behavior in gene expression across different genotypes and P treatments and identified DE circRNAs underlying the LP stress-related circRNAs. On the basis of the threshold ($|log2FC| > 1$ and $P < 0.05$), 120 of the 371 circRNAs were expressed at significantly different levels in four groups (including HP-NR-vs-LP-NR and HP-BR-vs-LP-BR, which denote comparisons between different P levels in the same genotype, and HP-NR-vs-HP-BR and LP_NR-vs-LP_BR, which denote comparisons between different genotypes at the same P level) (Table 1, S2 Table).

The relative expression levels of the 120 identified DE circRNAs were analyzed in each sample according to the RPM method (Fig 3A). It is clear that there were significant differences between the two genotypes. The circRNAs in the middle of the heat map may play a fundamental role in the roots of soybean plants under LP stress, while the genotype-specific circRNAs on either end of the heat map may be very important for soybean plants to respond to LP stress, which requires further study. In addition, a Venn diagram and histogram were constructed to indicate changes in expression that were specific and common among the four groups (Fig 3B and 3D). Briefly, the greatest amount of DE circRNAs occurred in the comparisons of the two soybean genotypes, NN94156 and Bogao, under both P conditions, especially under the LP conditions (Fig 3B and 3D). These results indicate that circRNAs exhibit genotype-specific regulatory patterns, which may play potential roles in tolerance to P deficiency.

In contrast, in the roots of NN94156 plants under LP stress, the expression of only three DE circRNAs was upregulated, and that of one circRNA was downregulated, while in the roots of the sensitive B plants, the expression of 15 and 13 circRNAs was upregulated and downregulated, respectively (Fig 3B and 3D). These results showed that the expression of more DE circRNAs changed significantly under LP stress in the sensitive genotype (Bogao) than in the tolerant genotype (NN94156), suggesting that the tolerant genotype was slightly affected by LP stress; in other words, compared with Bogao, NN94156 may have a better ability to maintain P

**Table 1. Statistical analysis of intergroup significantly DE circRNAs.**

| Comparison groups | Upregulated DE circRNAs | Downregulated DE circRNAs | Total DE circRNAs |
|---|---|---|---|
| HP-NR-vs-LP-NR | 3 | 1 | 4 |
| HP-BR-vs-HP-NR | 30 | 37 | 67 |
| HP-BR-vs-LP-BR | 15 | 13 | 28 |
| LP-BR-vs-LP-NR | 39 | 54 | 93 |
| Total | 87 | 105 | 192 |

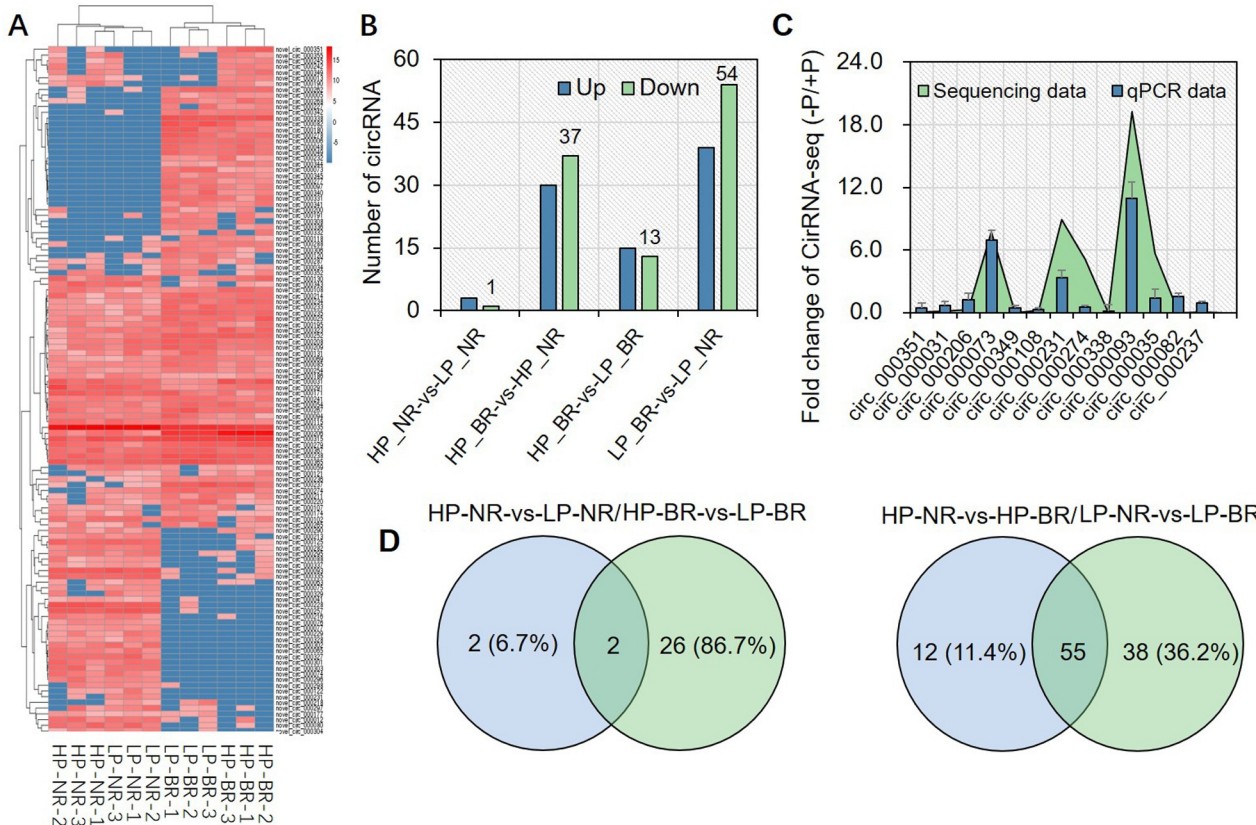

**Fig 3. DE circRNAs in the roots of soybean plant under LP stress.** (A) Heatmap and cluster analysis of circRNA expression levels in each sample. The expression levels were calculated according to log2FC values. The red/blue color indicates greater/lower levels of circRNA expression, respectively. (B) Number of DE circRNAs in each comparison. The numbers above the columns show the number of upregulated (blue) and downregulated (green) circRNAs. HP-NR-vs-LP-NR and HP-BR-vs-LP-BR denote comparisons between different P levels in the same genotype; HP-NR-vs-HP-BR and LP_NR-vs-LP_BR denote comparisons between different genotypes at the same P level. (C) Real-time PCR-based verification of the DE circRNAs on the basis of the RNA-seq data. (D) Venn diagrams of DE circRNAs identified in the different comparisons.

homeostasis under LP stress. In addition, the circRNAs uniquely DE in each genotype or under each P level were identified (S2 Table). These results revealed that the DE circRNAs may play an important role in the response to LP stress in soybean.

We randomly selected 13 circRNAs and designed divergent primers (S3 Table) to validate the results of the DE circRNAs via qPCR experiments. The qPCR results were highly correlated ($R^2$ = 0.82) with the RNA-seq results (Fig 3C), indicating the robustness of our study in the identification of circRNAs in response to P stress. To clarify the presence of the circRNAs further, a representative circRNA, novel_circ_000237, was selected, and a pair of divergent primers were designed. Both cDNA and gDNA (negative control) were then used as templates for reverse transcription RT-PCR amplification. The presence of the circRNAs and the reliability of the data were subsequently successfully validated by Sanger sequencing experiments (Fig 4, S2 Fig).

## Putative functions and expression patterns of DE circRNAs and their circRNA-host genes

Studies have shown that circRNAs play an important role in gene regulation through the *cis*-regulatory activity of its circRNA-host genes [43]. In this study, to analyze the potential

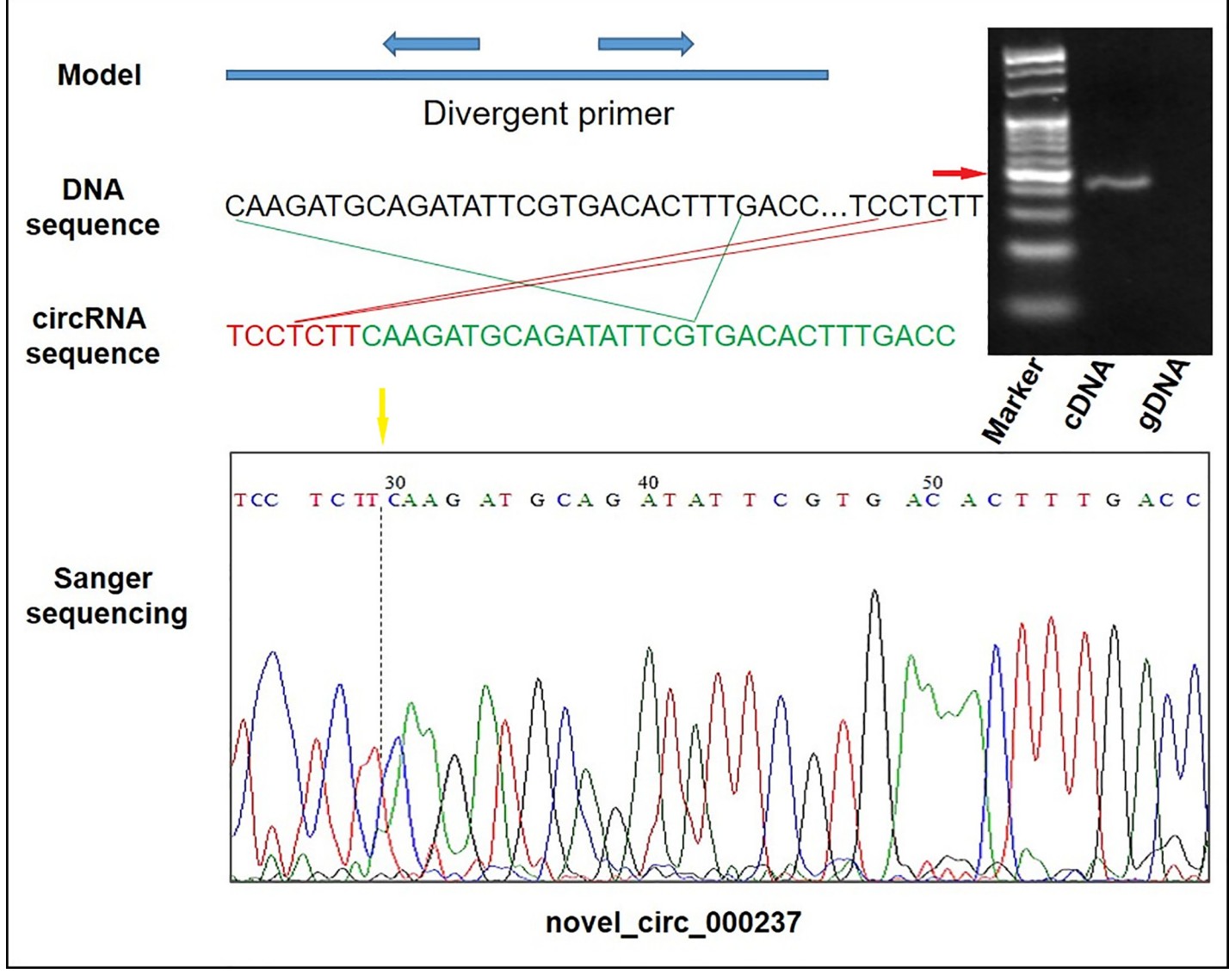

**Fig 4. Validation of a circRNAs (novel_circ_000237) by qPCR and Sanger sequencing.** M, DL500 marker and the red arrow represent 100 bp. The yellow arrows denote the divergent primers for PCR amplification orientation.

possible roles of DE circRNAs between the sensitive genotype Bogao and the tolerant genotype NN94156 under LP stress, we first predicted and annotated the circRNA-host genes of those DE circRNAs (S2 Table). The annotated genes that encode circRNAs were considered the cir-cRNA-host genes of circRNAs, and the intergenic-type circRNA fragments originating between two genes were designated as no circRNA-host genes (NAs). As a result, among the 120 DE circRNAs, 101 were identified originating from 84 circRNA-host genes, and 19 cir-cRNAs had no circRNA-host genes (S1 Table). Previous studies [2] have shown that the plant response to LP stress involves a complex regulatory network with many different biological processes involved, such as energy production, nucleic acid (DNA, RNA) synthesis, photosyn-thesis, glycolysis, respiration, cell membrane formation, redox reactions, signal transduction and nitrogen fixation [44]. In this study, GO enrichment analysis of the circRNA-host genes revealed that the enriched GO terms were related mainly to the defense response, ADP

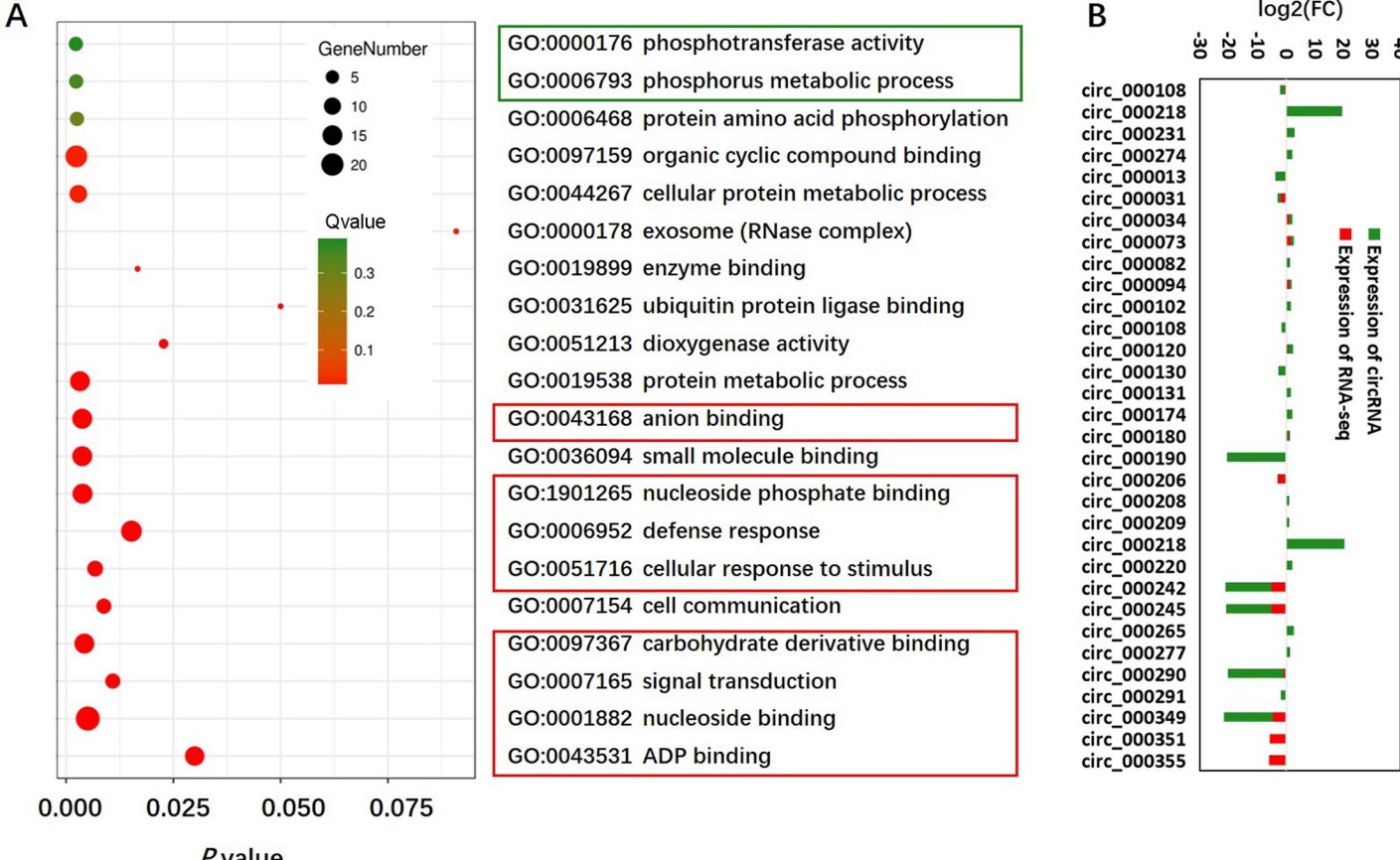

**Fig 5. GO enrichment analysis of DE circRNAs and comparisons between circRNA expression patterns and their circRNA-host genes.** (A) GO enrichment analysis of circRNA-host genes of DE circRNAs in response to LP stress. The red and green boxes highlight the gene clusters involved in response to abiotic stress and LP stress, respectively. (B) Similar expression patterns of circRNAs and their circRNA-host genes were defined as "positive". In detail, if the expression of a circRNA and that of its circRNA-host gene are both upregulated and downregulated after LP stress, the pattern was defined as "positive"; if not, "negative".

binding, nucleotide binding, organic substance catabolic processes, oxidoreductase activity, and signal transduction, suggesting the involvement of these circRNAs in the responsiveness of soybean to P deficiency (Fig 5A, S4 Table). Moreover, six circRNA-host genes were related to phosphate metabolic processes, although the associated GO terms were not significantly enriched.

Previous studies have shown that weak correlations exist between the expression of circRNAs and that of linear RNAs in both animal and human cells [38, 45]. In plants, however, studies have shown that the expression of circRNAs is positively correlated with that of their circRNA-host genes [29]. In the present study, we observed that some of the LP-responsive circRNAs were encoded by circRNA-host genes, which may play important roles under LP stress and exhibited differential expression patterns under LP stress (S2 Table). These observations revealed that the expression of circRNAs induced by LP stressed might be similar to the expression of their circRNA-host genes. We thus compared the differences in expression patterns between circRNAs and their circRNA-host genes by selecting circRNAs whose expression was significantly differentially up- or downregulated in each genotype after LP stress. If the expression patterns of a circRNA and its host gene indicated consistent upregulation or downregulation after LP stress, the circRNA was defined as a "positive" responsive circRNA;

otherwise, it was defined as "negative". Interestingly, with respect to the 32 DE circRNAs (four for HP-NR-vs-LP-NR and 28 for HP-BR-vs-LP-BR) (Table 1), a similar pattern was observed between the majority of circRNAs (28, 88%) and their circRNA-host genes under LP stress (Fig 5B). This finding suggested that the expression of most circRNAs was positively correlated with their host gene expression after LP stress in soybean.

## Putative functions of circRNA-mediated miRNA networks

By acting as miRNA sponges and competing for endogenous mRNAs, circRNAs play crucial roles in regulating functional gene expression and transcription [46]. Many miRNAs have been reported to participate in the response to P deficiency in various plant species by regulating the expression of target genes, including miR399 [47–50], miR319 [51], miR156 [52], and miR159 [53]. For example, a recent study in maize revealed that the PILNCR1-miR399 regulatory module is important for LP tolerance [47]. Similarly, Tian et al. [8] reported that miR399 functions as a potential integrator of phosphate homeostasis. The regulatory protein *pho2* results from a nonsense mutation in the target gene of miR399, which can cause excessive phosphate accumulation [50]. Overexpression of *GmMIR319* in tobacco improved tolerance to P deficiency [51]. Thus, in the present study, to reveal the functional importance of circRNAs further, we first identified all circRNA-originating target miRNAs (including circRNA-targeted miRNAs and miRNAs targeted by mRNAs). The candidate miRNA pairs were then screened on the basis of the DE circRNAs (S2 Table). We ultimately found that 70 out of 120 DE circRNAs contain miRNA-binding sites, which may function as miRNA sponges in response to P deficiency in soybean. Target prediction indicated that 570 miRNAs were targeted by these 70 circRNAs (S5 Table). Interestingly, we found that miR399, miR319, miR156 and miR159 were sponged by several DE circRNAs. Moreover, we also detected various novel DE circRNAs expressed specifically in the LP-sensitive genotype (Bogao) between the different P levels (HP-BR and LP-BR). For example, the novel circRNAs novel_circ_000013, novel_circ_000349, novel_circ_000351, and novel_circ_000277 acted as sponges of many miRNAs, suggesting that these novel DE circRNAs may play an important role in the response to LP stress (Fig 6, S6 Table), similar to how miR838 reportedly responds to abiotic stress [54]. These circRNAs may regulate their target genes in soybean in response to P deficiency by sponging these miRNA family members.

miRNAs are involved in various plant physiological processes in response to environmental stresses by regulating the expression of functional genes [55]. To reveal the potential roles of soybean circRNAs in response to LP stress further, the target mRNAs of miRNAs were predicted by bioinformatic analysis. As a result, 95 mRNAs were identified out of the 570 putative circRNA-binding miRNAs (S7 Table). To determine the gene functions associated with the DE circRNAs, 95 targeted mRNAs were subjected to GO analysis. Interestingly, similar to that revealed by the GO analysis of the circRNA-host genes, ADP binding (GO: 0043531), defense response (GO: 0006952), nucleotide binding (GO: 0001883), signal transduction (GO: 0007165), phosphorylation (GO: 0016310), and phosphate metabolic processes (GO: 0006796) were specifically enriched (S8 Table). The GO analysis results of the predicted mRNAs revealed that the targets of the DE circRNAs in response to LP stress are involved in various functions associated with different biological processes, cellular components, and molecular functions. We speculate that the expression of circRNAs may change in response to LP stress because the circRNAs act as negative regulators of their circRNA-host genes.

In addition, to analyze the function of these DE circRNAs further, we used the Kyoto Encyclopedia of Genes and Genomes (KEGG) pathway annotation method to analyze the predicted target mRNAs of circRNAs, and 21 pathways were revealed. These KEGG pathways are

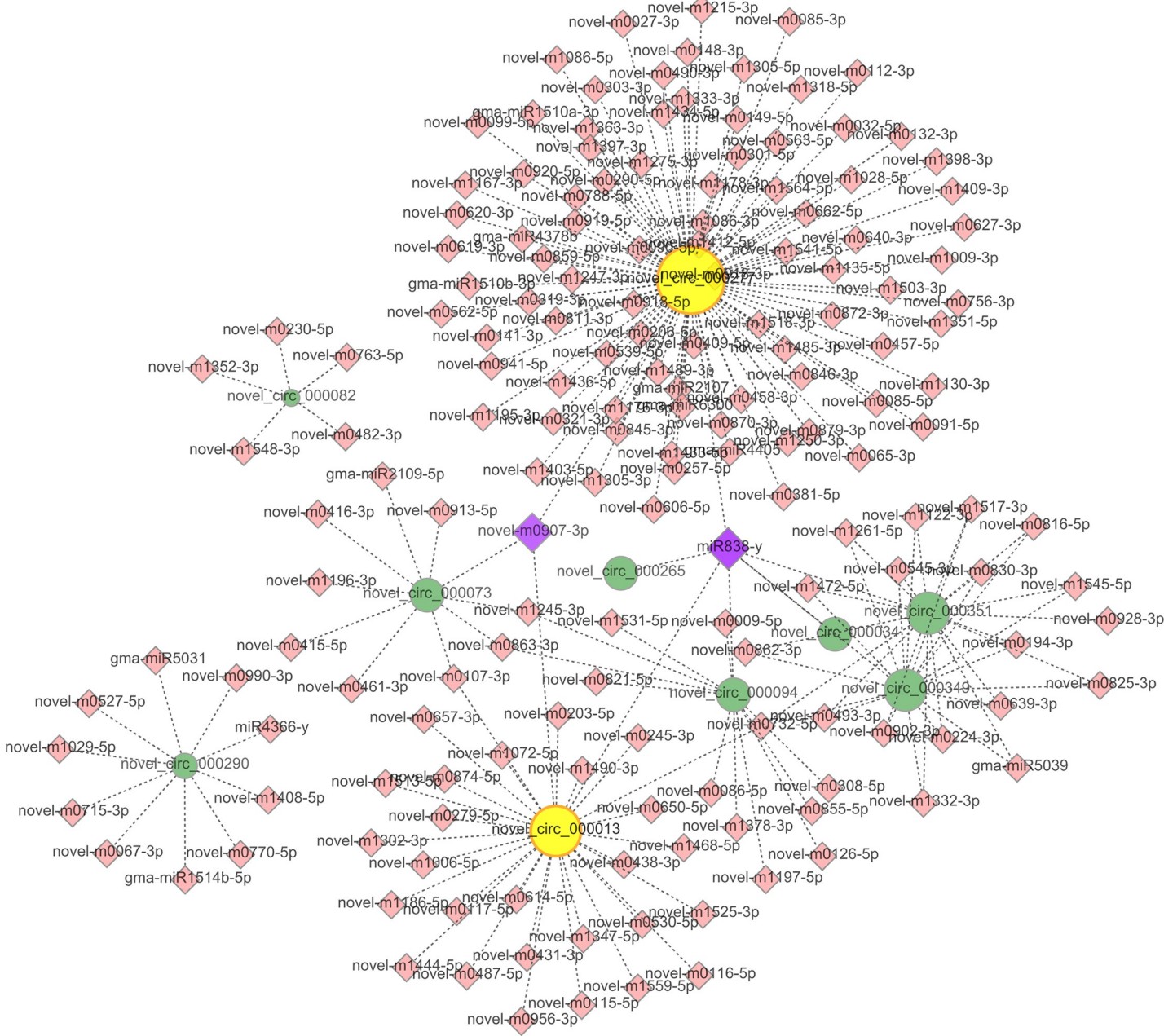

**Fig 6. Prediction of circRNA-associated-miRNA networks in response to LP stress.** The circRNA-miRNA and miRNA-gene interaction networks were constructed on the basis of the HP-BR vs LP-BR comparison. The circles denote circRNAs, the large circular areas denote miRNAs associated with a circRNA, the rhombuses denote miRNAs, and the purple rhombuses denote different miRNA-gene interactions.

involved mainly in the seleno-compound and thiamine metabolic pathways, plant-pathogen interactions, secondary metabolism, phenylpropanoid biosynthesis, and so forth (S9 Table). Previous studies have shown that plants absorb P, sulfur, and selenium in the form of anions from the soil. A deficiency in P may affect the absorption and accumulation of sulfur and selenium by plants [56, 57]. The nucleotide-binding site-leucine-rich repeat (NBS-LRR) gene family plays a crucial role in the disease resistance and various resistance responses of plants [58]. In the present study, 13 of the 95 target genes were NBS-LRR family genes (S7 Table),

suggesting that NBS-LRR family genes in soybean may play crucial roles in the response to LP stress and may be regulated by circRNAs involved in P metabolism. In addition, we found that many of the predicted targets of miRNAs encode transcription factors (TFs), including bHLH, WRKY, and nuclear TFs (S7 Table), which play crucial roles in plants under environmental stress [59]. For example, the predicted gene *Glyma.13G117700* is the target gene for novel_-circ_000232 and encodes WRKY TF 47, whose overexpression was proven to improve P-use efficiency in our subsequent experiments (unpublished). In addition, there are other predicted genes that may be related to P-use efficiency, such as the stress-induced protein SAM22 (*Glyma.17G030300*), the type IV inositol polyphosphate 5-phosphatase (*Glyma.07G107400*), and the TF bHLH51 (*Glyma.02G258900*). Overall, our results not only broaden the understanding of circRNAs in plants but also help elucidate the mechanism of circRNA responses to LP stress in soybean.

## Conclusions

Our results revealed that LP stress can significantly alter the genome-wide profiles of circRNAs. The treatment-specific and genotype-specific DE circRNAs identified in this study may be involved in plant responses to LP stress via posttranscriptional regulation of miRNA networks. Moreover, our results provide some clues for further investigation of the functions of circRNAs in plant responses to LP stress.

## Supporting information

**S1 Fig. Distribution of different length ranges of circRNAs (e.g., 500 represents 0–500).**
(TIF)

**S2 Fig. circRNA validation by RT-PCR and Sanger sequencing.** Thirteen circRNAs are shown in the figure. The successfully validated circRNAs included the following: 1, novel_-circ_000274; 3, novel_circ_000338; 5, novel_circ_000035; 6, novel_circ_000093; 8, novel_-circ_000108; and 11, novel_circ_000237. The unsuccessfully validated circRNAs included the following: 2, 4, 7, 9, 10, 12 and 13 (M, DL500 marker). Detailed information can be found in S3 Table.
(TIF)

**S1 Table. Detailed information of identified circRNAs in soybean roots.**
(XLSX)

**S2 Table. Significant DE circRNAs detected among the four comparison groups.**
(XLSX)

**S3 Table. Divergent primers used for circRNA validation.**
(XLSX)

**S4 Table. GO enrichment analysis of the circRNA-host genes of significantly DE circRNAs.**
(XLSX)

**S5 Table. miRNAs sponged by 70 DE circRNAs.**
(XLSX)

**S6 Table. miRNAs sponged by DE circRNAs in the HP-BR and LP-BR comparisons.**
(XLSX)

**S7 Table. miRNA-targeted mRNAs and their annotations.**
(XLSX)

**S8 Table. GO enrichment analysis of the targeted mRNAs of significantly DE circRNAs.**
(XLSX)

**S9 Table. KEGG pathway annotation of the predicted target mRNAs of circRNAs.**
(XLSX)

## Acknowledgments

We thank Dr. Zhenbin Hu from Kansas State University for his critical reading of the manuscript. Three anonymous reviewers are thanked for their critical and highly valued comments.

## Author Contributions

**Conceptualization:** Dan Zhang.

**Data curation:** Dan Zhang.

**Formal analysis:** Lingling Lv.

**Funding acquisition:** Haiyan Lü, Dan Zhang.

**Investigation:** Lingling Lv, Kaiye Yu, Xiangqian Zhang, Xiaoqian Liu.

**Methodology:** Lingling Lv, Kaiye Yu.

**Resources:** Xiaohui He.

**Software:** Haiyan Lü, Xiangqian Zhang.

**Supervision:** Dan Zhang.

**Validation:** Lingling Lv, Chongyuan Sun.

**Visualization:** Xiaoqian Liu, Huanqing Xu.

**Writing – original draft:** Lingling Lv.

**Writing – review & editing:** Haiyan Lü, Jinyu Zhang, Xiaohui He, Dan Zhang.

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
