## [Decision Letter · Decision Letter 0]

14 Oct 2019

PONE-D-19-24444

Transcriptome-wide identification of novel circular RNAs in soybean response to low-phosphorus stress

PLOS ONE

Dear Dr Zhang,

Thank you for submitting your manuscript to PLOS ONE. After careful consideration, we feel that it has merit but does not fully meet PLOS ONE’s publication criteria as it currently stands. Therefore, we invite you to submit a revised version of the manuscript that addresses the points raised during the review process.

We would appreciate receiving your revised manuscript by Nov 28 2019 11:59PM. To enhance the reproducibility of your results, we recommend that if applicable you deposit your laboratory protocols in protocols.io, where a protocol can be assigned its own identifier (DOI) such that it can be cited independently in the future. For instructions see: http://journals.plos.org/plosone/s/submission-guidelines#loc-laboratory-protocols

We look forward to receiving your revised manuscript.

Kind regards,

Zhi Min Yang

Academic Editor

PLOS ONE

Journal Requirements:

2. Please ensure that you refer to Figure 4 in your text as, if accepted, production will need this reference to link the reader to the figure.

3.  Thank you for stating the following in your Competing Interests section: NO

4.  Thank you for stating the following financial disclosure: NO

a) Please provide an amended Funding Statement that declares *all* the funding or sources of support received during this specific study (whether external or internal to your organization) as detailed online in our guide for authors at http://journals.plos.org/plosone/s/submit-now.  

b) Please state what role the funders took in the study.  If any authors received a salary from any of your funders, please state which authors and which funder. If the funders had no role, please state: "The funders had no role in study design, data collection and analysis, decision to publish, or preparation of the manuscript."

Reviewers' comments:

Reviewer's Responses to Questions

**Comments to the Author**

1. Is the manuscript technically sound, and do the data support the conclusions?

Reviewer #1: Yes

Reviewer #2: Yes

2. Has the statistical analysis been performed appropriately and rigorously? 

Reviewer #1: Yes

Reviewer #2: Yes

3. Have the authors made all data underlying the findings in their manuscript fully available?

Reviewer #1: No

Reviewer #2: Yes

4. Is the manuscript presented in an intelligible fashion and written in standard English?

Reviewer #1: Yes

Reviewer #2: No

5. Review Comments to the Author

Reviewer #1: 'Transcriptome‑wide identification of novel circular RNAs in soybean response to low-phosphorus stress' is a good paper with some new findings for the identification of circRNAs associated with soybean responses to low-P stress. The experiments are planned and conducted sensibly. In general the data are well described and well analyzed. This manuscript revealed that low-P stress can alter genome-wide profiles of circRNAs significantly. Then presented some treatment-specific and genotype-specific DE circRNAs, which involve in plant responses to low-P stress and provided some clues for further investigating the functions of circRNAs in plant responses to low-P stress. I prefer to accept the paper to be published in this journal with minor revision and mirror grammar editing (suggest to use a professional company).

Minor points

1. The discussion section can be more informative and in-depth, such as what potential functions those candidate circRNAs may have compared with previous studies.

2. Have the authors made the data publicly available? I cannot find any reference to this.

Reviewer #2: The manuscript entitled 'Transcriptome-wide identification of novel circular RNAs in soybean response to low-phosphorus stress' described the profiling of genome-wide circular RNAs of low-phosphorus-treated soybean by RNA-sequencing. Many previous-unknown circular RNAs were identified and some are specifically responsive to low-phosphorus stress. This study provided the fundamental information about the circular RNA profiling in soybean and many interesting candidates for further investigations of soybean response to low-phosphorus stress. However, some major and minor issues should be addressed before reconsideration for publication.

major point:

Although 13 randomly-selected circRNAs were validated on their expression by qRT-PCR, only one was validated on its existance and reliability by divergent PCR and Sanger sequencing. The previous studies showed that RNA-sequencing results are not 100% reliable. Therefore, please check the reliability of rest 12 circRNAs by divergent PCR and Sanger sequencing.

minor points:

1. Page 3 Line 43: "in carbon metabolism an in regulation and signaling". Please check the spelling.

2. Figure 1B and Page 10 Line 217: Please give the full name of "annot_exons" in the figure legend. I think "multiple_exons" is more appropriate than "annot_exons", considering that there is a type of one_exon.

3. Page 10 Line 220 "there is a close relationship...": please explain what kind of relationship it is.

4. Fig 5A: there are two "P-value" indicators (One is the color, the other is the axis X) in the figure. Which one is right?

5. Page 16 Line 350-353 "Previous studies shown ...nitrogen fixation": The sentence has obvious grammar problems.

6. In the manuscript, the word "involve" is not grammatically used. Please also check the whole manuscript for other grammar issues.

7. Page 20 Line 445: there are two "of" in the sentence.

6. PLOS authors have the option to publish the peer review history of their article (what does this mean?). If published, this will include your full peer review and any attached files.

Reviewer #1: No

Reviewer #2: No

---

## [Author Response · Author response to Decision Letter 0]

27 Nov 2019

Dear Prof. Yang,

We appreciate the time you and the reviewers spent reviewing our manuscript entitled ‘Transcriptome-wide identification of novel circular RNAs in soybean in response to low-phosphorus stress’ (PONE-D-19-24444)' at PLOS ONE and your insightful comments and suggestions, which have helped us improve our manuscript.

After we received the comments, we immediately designed new experiments (we validated the circRNAs by divergent PCR and Sanger sequencing) and carefully revised the manuscript as suggested. We used a language editing agency to edit the manuscript such that formal academic language was used and that each sentence is clear and has been accurately delivered. Along with the revised manuscript, we also submit our point-by-point responses to the reviewers’ comments. The revisions are marked in red in the revised manuscript. With the help of your excellent suggestions, we believe that our revised manuscript has undergone significant and substantial improvement.

Thank you again for your consideration, and we hope that the revised manuscript is publishable in PLOS ONE.

Sincerely,

Dan Zhang

Below are our point-by-point responses:

In this study, authors performed the genome-wide identification of circular RNAs in soybean, and then profiling their expression patterns, detecting the differentially expressed ones, carried out GO annotation and enrichment, and finally experimentally confirmed selected ones. It is meaningful and interesting to readers. Generally speaking, the work was well designed and the result was sufficient to support their conclusion. Meanwhile, the writing and description are OK. However, I have a few comments that should be addressed to ensure the improving of the manuscript.

Major points:

The paper of Zhao et al. 2017. Scientific report. Also reported the circRNAs in soybean, which should be cited and discussed by authors. The previously results reported by Zhao et al (2017) and the results reported by authors should be compared and discussed, including the number of detected circRNAs, the same and novel circRNAs between the two studies, the circRNAs identification procedures. Because these are important to fully understand the circRNAs reported by authors in this paper. Moreover, several recently published article related to circRNAs in plants under abiotic stress including cucumber should be mentioned and maybe discussed in the article (Zhu et al 2019, BMC, Identification of cucumber circular RNAs responsive to salt stress).

Response: This is a great point that has made our discussion more interesting. We added some points from recently published papers (Zhao et al. 2017; Zhu et al 2019) to the Discussion section.

Comparisons have shown that circRNAs are expressed specifically in the leaves, stems and roots, indicating that circRNAs are tissue specific (Zhu et al 2019). Moreover, we also discussed the possible reason for the lower number of circRNAs in our study, which might be P-supply specific. In addition, because the circRNA identification procedures and the names in these two studies are different, there was no comparison of the same and novel circRNAs between the two studies.

Zhu Y X , Jia J H , Yang L , et al. Identification of cucumber circular RNAs responsive to salt stress. BMC Plant Biology, 2019, 19(1).

Zhao W, Cheng YH, Zhang C, You QB, Shen XJ, Guo W, et al. Genome-wide identification and characterization of circular RNAs by high throughput sequencing in soybean. Sci Rep. 2017; 7. doi: 10.1038/s41598-017-05922-9.

L159. Previously, gene was defined as DNA fragments coding proteins. With the advance of knowledge, besides coding genes, numerous non-coding genes, including miRNA, lncRNA, circRNA and so on, were reported. It is similar in soybean. 

I suggest authors carefully used the word “source gene”, as from your description, it apparently only represents the coding genes. Meanwhile, I suggest that authors should analysis which circRNA origin from lncRNA to further expand the source genomic region of circRNA.

Response: Thank you for the suggestions. We have added a corresponding description and the “source genes” to “circRNA-host genes” in the revised manuscript. Regarding lncRNA, the library construction method was the same as that for circRNA. We are already preparing manuscripts specifically related to lncRNA.

L151. It is OK that the circRNA “were annotated by blasted against the circBase”. But the parameters were not given. Furthermore, how to make sure those annotated circRNA are not novel circRNAs, as those well-matched hits also can be different circRNAs, although they shared same annotation based on BLAST analysis.

Response: The identification of circRNAs according to the E value of the comparisons were as follows: circRNAs were searched against circBase via BLAST to determine their annotations. An E value < e-10 was used as the filtering criterion, and the circRNAs meeting this alignment criterion were defined as existing circRNAs; conversely, they were defined as newly predicted circRNAs. The related discussion has been added to the Materials and Methods section (page 7, lines 154-156).

L228. Authors said that “indicating that the circRNAs were generated by various mechanisms”. It is much unclear. Short and long circRNAs and uneven distribution are the result of which kind of mechanisms? Please clarify.

Response: Thank you for this suggestion. We deleted the inappropriate statements.

Minor points:

L38. Change the keywords, as some of them had appeared in the title.

Response: This has been changed as suggested. Thanks.

L55. Change “is” to “are”.

Response: This has been changed as suggested. Thanks.

L77. Change “or” to “and”.

Response: This has been changed as suggested. Thanks.

L110. It should be Orthographic, not Italic, in “µM”. Change thoroughly.

Response: This has been corrected. Thanks.

L111. In Hoagland's, it should be Hoagland’s.

Response: This has been corrected. Thanks.

L115. Missing blank between 80 and °C. Please check thoroughly to avoid the missing blank between number and unit.

Response: We are especially thankful for your careful comments. We have added the spaced and have checked all of them thoroughly.

L135. Change “200-500” to “200 to 500”.

Response: This has been changed as suggested. Thanks.

L141. Remove the “TM” in HiSeqTM 4000, or make it superscript.

Response: This has been deleted. Thanks.

L146. Rewrite the sentence “The reads were filtered were based on three rules”.

Response: Thank you for this suggestion. We have changed the sentence “The reads were filtered were based on three rules” to “The reads were filtered according to three criteria” in the revised manuscript.

L148. Missing blank before and after “≤”.

Response: The space has been added. Thanks.

L146-148. As words after (1) and (2) are complete sentence, the (3) also should sentences, (3) should also be complete sentences to match the same pattern.

Response: This has been corrected as suggested. Thanks.

L174. Change the form of “2(-△△CT)”. Usually, it shows as 2-△△Ct.

Response: This has been corrected. Thanks.

L149 and L197. Keep the same format of “Wm82.a2.v1”. In L149, it is Italic; in L197, it is not.

Response: This has been corrected. Thanks.

L214. Remove “an”. As it is one to more, not always one exon.

Response: This has been deleted. Thanks.

L287. Make sure there is blank before and after “>” and thoroughly check all over the paper.

Response: We have added the necessary spaces and have checked all of them thoroughly.

L343. The “cis” should be Italic.

Response: This has been corrected.

L379. Add “” to the word negative to keep the same pattern as “positive”.

Response: These have been added.

L381. As you divided circRNAs into “positive” and “negative”. Please show how many circRNAs are “positive” and how many are “negative” in the most circRNAs (88%).

Response: These number have been added.

L401. Dose these 570 miRNAs belong to soybean. If it is, where and how did you collect them? Please clarify.

Response: Yes, these 570 miRNAs were from soybean. We collected them on the basis of the small RNA data, which were collected via an experiment whose design was the same as that of the circRNA experiment in this study, and relevant manuscripts are being prepared.

L404. Please give several examples of “specific-expression novel DE circRNAs” and show circRNAs target to which miRNA.

Response: We have added several examples of novel DE circRNAs whose expression is specific and have added information concerning the miRNA targets of circRNAs to the revised manuscript (page 18, lines 412-416).

L418-L419. Please convince me that the 570 miRNAs were only targeting to 95 mRNAs.

Response: In this study, we used Patmatch (v1.2) to predict mRNAs targeted by miRNA sponges and the interactions between the mRNAs and miRNAs. As shown in Table S7, the 570 miRNAs targeted only 95 mRNAs mainly because many miRNAs share a common target mRNA. As reported in many studies (Sun et al., 1996; Aftabuddin et al., 2014; Calderari et al., 2017), one miRNA can target hundreds of downstream target mRNAs, and an mRNA can be targeted by multiple miRNAs.

Sun H, Wakizaka Y, Rao AS, et al. Use of MHC class I or II "knock out" mice to delineate the role of these molecules in acceptance/rejection of xenografts. 1996, 28(2):732.

Aftabuddin M, Chittabrata Mal, Arindam Deb, et al. C2Analyzer: Co-target–Co-function Analyzer. Genomics, Proteomics & Bioinformatics, 2014, 12(3):133-136.

Calderari S D, Malika R, Garaud A, et al. Biological roles of microRNAs in the control of insulin secretion and action. Physiological Genomics, 2017, 49(1):1-10.

Reviewer #1: 'Transcriptome wide identification of novel circular RNAs in soybean response to low-phosphorus stress' is a good paper with some new findings for the identification of circRNAs associated with soybean responses to low-P stress. The experiments are planned and conducted sensibly. In general the data are well described and well analyzed. This manuscript revealed that low-P stress can alter genome-wide profiles of circRNAs significantly. Then presented some treatment-specific and genotype-specific DE circRNAs, which involve in plant responses to low-P stress and provided some clues for further investigating the functions of circRNAs in plant responses to low-P stress. I prefer to accept the paper to be published in this journal with minor revision and mirror grammar editing (suggest to use a professional company).

Minor points

1. The discussion section can be more informative and in-depth, such as what potential functions those candidate circRNAs may have compared with previous studies.

Response: Thank you for the suggestions. We added some points from recently published papers (Zhao et al., 2017; Zhu et al., 2019) to the Results and Discussion section (Page 20, lines 454-459). For example, the predicted gene Glyma.13G117700 is the target gene for novel_circ_000232 and encodes WRKY TF 47, whose overexpression was proven to improve P-use efficiency in our subsequent experiments (unpublished). In addition, there are other predicted genes that may be related to P-use efficiency, such as the stress-induced protein SAM22 (Glyma.17G030300), the type IV inositol polyphosphate 5-phosphatase (Glyma.07G107400), and the TF bHLH51 (Glyma.02G258900).

Zhu Y X , Jia J H , Yang L , et al. Identification of cucumber circular RNAs responsive to salt stress. BMC Plant Biology, 2019, 19(1).

Zhao W, Cheng YH, Zhang C, You QB, Shen XJ, Guo W, et al. Genome-wide identification and characterization of circular RNAs by high throughput sequencing in soybean. Sci Rep. 2017; 7. doi: 10.1038/s41598-017-05922-9.

2. Have the authors made the data publicly available? I cannot find any reference to this.

Response: We will make the data publicly available upon acceptance.

Reviewer #2: The manuscript entitled 'Transcriptome-wide identification of novel circular RNAs in soybean response to low-phosphorus stress' described the profiling of genome-wide circular RNAs of low-phosphorus-treated soybean by RNA-sequencing. Many previous-unknown circular RNAs were identified and some are specifically responsive to low-phosphorus stress. This study provided the fundamental information about the circular RNA profiling in soybean and many interesting candidates for further investigations of soybean response to low-phosphorus stress. However, some major and minor issues should be addressed before reconsideration for publication.

Major point:

Although 13 randomly-selected circRNAs were validated on their expression by qRT-PCR, only one was validated on its existance and reliability by divergent PCR and Sanger sequencing. The previous studies showed that RNA-sequencing results are not 100% reliable. Therefore, please check the reliability of rest 12 circRNAs by divergent PCR and Sanger sequencing.

Response: This is a great point that would make our results more credible. As suggested, we have checked the reliability of the 13 circRNAs, including the remaining 12 circRNAs, by divergent PCR and Sanger sequencing. The existence and reliability of six were validated by divergent PCR and Sanger sequencing (Fig S2 and Table S3). We added these results to the Results section of the revised manuscript (page 27, lines 636-641).

Minor points:

1. Page 3 Line 43: "in carbon metabolism an in regulation and signaling". Please check the spelling.

Response: This has been corrected. Thanks.

2. Figure 1B and Page 10 Line 217: Please give the full name of "annot_exons" in the figure legend. I think "multiple_exons" is more appropriate than "annot_exons", considering that there is a type of one_exon.

Response: Thank you for the suggestions. We have changed “annot_exons” to “multiple_exons” accordingly.

3. Page 10 Line 220 "there is a close relationship...": please explain what kind of relationship it is.

Response: We found that 147 of the 225 circRNAs produced two or more circRNAs, suggesting that the same gene can produce multiple circRNAs by alternative splicing. Therefore, we speculate that there is an intimate relationship between the origin mechanism of circRNAs and the splicing mechanism of precursor mRNAs, which is consistent with previous reports. We have added a corresponding explanation to the revised manuscript.

4. Fig 5A: there are two "P-value" indicators (One is the color, the other is the axis X) in the figure. Which one is right?

Response: We apologize for the mistake in Fig 5A. We have changed “color P-value” to “Q-value” in the revised manuscript.

5. Page 16 Line 350-353 "Previous studies shown ...nitrogen fixation": The sentence has obvious grammar problems.

Response: Thank you for this comment. We have reworded this sentence to “Previous studies have shown that the plant response to LP stress involves a complex regulatory network with many different biological processes involved, such as energy production, nucleic acid (DNA, RNA) synthesis, photosynthesis, glycolysis, respiration, cell membrane formation, redox reactions, signal transduction and nitrogen fixation”.

6. In the manuscript, the word "involve" is not grammatically used. Please also check the whole manuscript for other grammar issues.

Response: Thank you for this suggestion. The main text has been edited by the language editing agency American Journal Experts (http://www.aje.com/), and an editorial certification has been submitted along with the manuscript.

7. Page 20 Line 445: there are two "of" in the sentence.

Response: This has been corrected. Thanks.

---

## [Decision Letter · Decision Letter 1]

17 Dec 2019

Transcriptome-wide identification of novel circular RNAs in soybean in response to low-phosphorus stress

PONE-D-19-24444R1

Dear Dr. Zhang,

We are pleased to inform you that your manuscript has been judged scientifically suitable for publication and will be formally accepted for publication once it complies with all outstanding technical requirements.

With kind regards,

Zhi Min Yang

Academic Editor

PLOS ONE

Additional Editor Comments (optional):

Reviewers' comments:

Reviewer's Responses to Questions

**Comments to the Author**

1. If the authors have adequately addressed your comments raised in a previous round of review and you feel that this manuscript is now acceptable for publication, you may indicate that here to bypass the “Comments to the Author” section, enter your conflict of interest statement in the “Confidential to Editor” section, and submit your "Accept" recommendation.

Reviewer #1: All comments have been addressed

Reviewer #2: All comments have been addressed

2. Is the manuscript technically sound, and do the data support the conclusions?

Reviewer #1: Yes

Reviewer #2: Yes

3. Has the statistical analysis been performed appropriately and rigorously? 

Reviewer #1: Yes

Reviewer #2: Yes

4. Have the authors made all data underlying the findings in their manuscript fully available?

Reviewer #1: Yes

Reviewer #2: Yes

5. Is the manuscript presented in an intelligible fashion and written in standard English?

Reviewer #1: Yes

Reviewer #2: Yes

6. Review Comments to the Author

Reviewer #1: (No Response)

Reviewer #2: The authors performed the additional experiments I asked and addressed all my concerns appropriately. I recommended to accept the paper for publication.

7. PLOS authors have the option to publish the peer review history of their article (what does this mean?). If published, this will include your full peer review and any attached files.

Reviewer #1: No

Reviewer #2: No

---

## [Editor Report · Acceptance letter]

8 Jan 2020

PONE-D-19-24444R1 

Transcriptome-wide identification of novel circular RNAs in soybean in response to low-phosphorus stress 

Dear Dr. Zhang:

I am pleased to inform you that your manuscript has been deemed suitable for publication in PLOS ONE. Congratulations! Your manuscript is now with our production department. 

With kind regards,

on behalf of

Dr. Zhi Min Yang 

Academic Editor

PLOS ONE